- A satellite observation-based analysis of the
- 2 distribution and formation mechanism of ice crystal
- 3 number concentration over the Tibetan Plateau

Kai Wang<sup>1</sup>, Xiaocong Wang<sup>2</sup>, Qianshan He<sup>3</sup>, Hong Nie<sup>4</sup>, Yanyu

- Wang<sup>5</sup>, and Yonghang Chen<sup>6</sup>
- <sup>1</sup>College of Atmospheric Science, Nanjing University of Information Science and
- Technology, Nanjing, China
- <sup>9</sup> Institute of Atmospheric Physics, Chinese Academy of Sciences, Beijing, China
- <sup>3</sup>Shanghai Meteorological Bureau, Shanghai, China
- <sup>4</sup>Qinghai Meteorological Service Centre, Xining, China
- State Environmental Protection Key Laboratory of Formation and Prevention of
- Urban Air Pollution Complex, Shanghai Academy of Environmental Sciences,
- Shanghai, China
- <sup>6</sup>Institute of Desert Meteorology, China Meteorological Administration, Urumqi,
- China

17 Correspondence to: Qianshan He (oxeye75@163.com)

Abstract.

- Cirrus clouds are located at the upper middle-lower troposphere and play an important
- role in the Earth's energy balance and the atmospheric water cycle. This study utilizes
- DARDAR-Nice data within June to August from 2006 to 2016 (except 2011),
- combined with CloudSat cloud products and other related aerosol products, to analyze
- the distribution characteristics and formation mechanisms of ice crystal number
- concentration  $(N_i)$  in cirrus clouds over the Tibetan Plateau (TP). The results indicate
- that  $N_i$  over the northern TP is significantly lower than that over the southern region, mainly due to differences in underlying aerosol concentration and the intensity of
- mainly due to differences in underlying aerosol concentration and the intensity of convective activity. Dominated by homogeneous nucleation,  $N_i$  exhibits a typical 'V'
- 29 shaped vertical profile over the TP. When deep convective activity occurs, it
- 30 facilitates the increase in  $N_i$ . In contrast, dust and smoke aerosols hinder the formation

1

https://doi.org/10.5194/egusphere-2025-4514 Preprint. Discussion started: 15 October 2025 © Author(s) 2025. CC BY 4.0 License.

of  $N_i$  through heterogeneous nucleation.. Additionally, the vertical wind velocity near 400 hPa in the northern TP approaches zero, causing the  $N_i$  peak to appear prematurely below the homogeneous nucleation threshold temperature (-38 °C).

### 1 Introduction

Cloud is the key link in the energy and water vapor balance of the earth-atmosphere system and plays an important role in global weather and climate change (Wang and Zhao, 1994; Stephens, 2005). Cirrus clouds are composed of a large number of non-spherical ice crystal particles with a wide range coverage of Earth surface (Baran, 2012; Guignard et al., 2011), which can reflect solar short-wave radiation and absorb terrestrial long-wave radiation, affect the energy balance of the upper troposphere and stratosphere and play an important role in the global water cycle and climate change (Kienast-Sjögren et al., 2016). A definite knowledge of cirrus microphysical properties and their formation mechanism is an important prerequisite for deepening the understanding of global climate change.

The Tibetan Plateau (TP) is a highest and largest plateau of the world, known as the 'roof of the world', affects significantly the climate patterns in eastern and southwestern China, and even global, as well as the global water circulation system, due to the unique dynamic and thermal effects. In summer, South Asian high controls the TP, where the cirrus clouds show different characteristics from that in other regions along the same latitude. On the one hand, the TP and its southern slope serves as an important windows for troposphere-stratosphere material exchange, where the frequent deep convective activities in summer have transported water vapor and anthropogenic aerosol pollutants to the upper troposphere-lower stratosphere (UTLS) (Chen et al., 2019; Fu et al., 2006; Randel et al., 2010). On the other hand, the substantial elevation difference over the southern part of the TP and the topographic uplift movement promote warm and humid airmass rising into the upper troposphere, which is conducive to the occurrence and development of cirrus clouds (Zhao et al., 2019; Yang et al., 2020). Also, the accumulation of aerosols is conducive to the formation of cirrus ice crystals by heterogeneous nucleation.

https://doi.org/10.5194/egusphere-2025-4514 Preprint. Discussion started: 15 October 2025 © Author(s) 2025. CC BY 4.0 License.

1 So far, the study on cirrus clouds over the TP mainly focused on the spatiotemporal variation characteristics, cloud height, and cirrus cloud formation 2 mechanism. Xue et al. (2018) found that the occurrence frequence, average effective 3 4 radius of ice particles and cloud top height reached the maximum in summer over the TP using Moderate-Resolution Imaging Spectroradiometer (MODIS). Gao et al. (2003) 5 found that the occurrence frequency of cirrus clouds reached a maximum in April and 6 7 a minimum in November by MODIS data. Chen and Liu (2005) found that the occurrence of cirrus clouds over the TP in March and April was closely related to the 8 slow uplift of warm and humid airmass to the tropopause due to topographic effect. Li 9 et al. (2005) found that deep convection activities affected by Asian Summer 10 Monsoon (ASM) were closely related to cirrus cloud formation over the TP using 11 satellite observations. Zhang et al. (2020) used Cloud-Aerosol Lidar and Infrared 12 Pathfinder Satellite Observations (CALIPSO) to investigate the generation 13 14 mechanism of plateau cirrus clouds, revealing that large-scale orographic uplift, temperature fluctuations, and deep convection play crucial roles in their formation. 15 16 Documented research suggests that the formation of cirrus ice crystal particles is 17 at the mercy of three mainstream mechanisms: deep convective cloud anvil overflow (Prabhakara et al., 1993; Wang et al., 1996), homogeneous nucleation and 18 19 heterogeneous nucleation (wang et al., 1997; Cantrell and Heymsfield, 2005; Chen et 20 al., 2000). Updrafts and strong horizontal currents in the upper troposphere induced by deep convective activities lead to rapidly spreading around into cloud anvils 21 composed of ice crystal particles (Takahashi and Luo, 2012). The homogeneous 22 23 nucleation of water vapor to form ice crystals requires a temperature below -38 °C and high relative humidity. While heterogeneous nucleation to form ice crystals 24 25 requires relatively higher ambient temperature but insoluble aerosol particles (such as black carbon, dust) as ice-nucleating particles (INPs) (Morris et al., 2004; Murray et 26 27 al., 2010; Fan et al., 2019; Shi et al., 2015). Different formation mechanisms result in different effects on the microphysical 28 characteristics of cirrus ice crystals, in which ice crystal number concentration  $(N_i)$ 29 plays a crucial role in understanding and characterizing cirrus clouds (Comstock et al., 30

https://doi.org/10.5194/egusphere-2025-4514 Preprint. Discussion started: 15 October 2025 © Author(s) 2025. CC BY 4.0 License.

1

2 3

4 et al., 2000; Kay and Wood, 2008; Hendricks et al., 2011). However, current climate models and satellite observations face significant limitations in obtaining and utilizing 5  $N_i$ , which can lead to substantial biases in simulating cloud microphysical processes, 6 7 evaluating aerosol-cloud interactions, and calculating indirect radiative effects (Zhang et al., 2013; Sourdeval et al., 2018). It is supposed that homogeneous nucleation is a 8 dominant mechanism to decide  $N_i$  (Cantrell and Heymsfield, 2005). When enough 9 INPs occurs in the atmosphere, heterogeneous nucleation precedes homogeneous 10 nucleation to form ice crystals, resulting in a consumption of a large amount of water 11 12 vapor and a decrease in the ambient supersaturation. Suppressed homogeneous nucleation will further impede the increase in the N<sub>i</sub> (Chen et al., 2000; Kärcher and 13 14 Lohmann, 2003; Shi et al., 2017). In the region of less convective activities, the effective radius of ice particles increases with the increase of INPs. Jin et al. (2007) 15 used a three-dimensional storm cloud model (IAP-CSM3D) to analyze the relation 16 17 between convective activity and cirrus cloud, found that the number concentration of 18 ice crystal formed by deep convective cloud anvil overflow decreases with a decrease 19 in water vapor from the convective activity transports. 20 However, the 3D-distribution characteristics of  $N_i$  and the corresponding contribution from deep convective cloud anvil overflow, homogeneous nucleation and 21 heterogeneous nucleation over the TP is not very clear. This study uses 22 23 liDAR-raDAR-Number concentration of ICE particles (DARDAR-Nice) data to analyze the spatial distribution characteristics of medium-upper cirrus clouds in 24 summer from 2006 to 2016 (except 2011), over the TP. The formation mechanism of 25 ice particles also be explored in combination of CALIPSO satellite aerosol products 26 with reanalysis data. Furthermore, this also sheds light on the role of aerosols in the 27 upper atmosphere of the TP in the process of cirrus ice crystal formation. The results 28 29 will contribute to a deeper understanding of the thermodynamic effects of the TP and further improve the accuracy of climate simulations. 30

2008). N<sub>i</sub> is widely used as a key variable in cloud forecasting to predict cloud evolution and is potentially closely linked to aerosol concentrations, making it an

important indicator for studying the impact of aerosols on ice cloud formation (Khain

\_\_

1

3

### 2 Data and methods

### 2.1 Satellite observations

4 This study uses ten summers of multi-satellite observations during 2006 to 2016, 5 except for 2011 due to data gaps, to investigate the distribution characteristics and formation mechanism of ice crystal particles over the TP. The primary dataset is the 6 DARDAR-Nice product, complemented by additional retrievals from CloudSat and 7 8 CALIPSO observations. 9 The DARDAR-Nice PRO product provides high-vertical-resolution estimates of 10  $N_i$  retrieved along the A-Train satellite track. The retrievals are based on the VarCloud algorithm (Delanoë and Hogan, 2008; 2010), which combines observations from the 11 Cloud Profiling Radar onboard CloudSat and the cloud-aerosol Lidar with Orthogonal 12 13 Polarization (CALIOP) lidar on CALIPSO. DARDAR-Nice profiles are provided with a vertical resolution of 60 m. The product includes  $N_i$  values and corresponding 14 15 uncertainty estimates for particle with size larger than 5, 25 and 100 µm. This production has been systematically and comprehensively evaluated based on 16 theoretical considerations and a large body of in situ observations (Sourdeval et al., 17 2018). However, it tends to overestimate ice crystal number concentrations in cloud 18 parcels warmer than -30 °C, due to the assumption of a monomodal particle size 19 distribution in the retrieval algorithm. To ensure the reliability of the results, this study 20 focuses exclusively on clouds with temperatures below -30°C and discusses the  $N_i$  of 21 22 ice crystals with sizes larger than 5 μm. In addition, although ice water content (IWC) is also provided in the 23 DARDAR-Nice product, it has not been specifically validated. Therefore, this study 24 25 uses **IWC** data from the CloudSat 2B-CWC-RO product 26 (ftp://ftp.cloudsat.cira.colostate.edu/), for which the retrieval quality and accuracy 27 have been discussed in detail by Austin et al. (2007, 2009). Besides 2B-CWC-RO, the 28 CloudSat 2B-CLDCLASS-LIDAR product is employed, which classifies clouds into eight types across ten vertical layers with a horizontal resolution of 2.5 km × 1.4 km. 29 Its classification algorithm integrates vertical and horizontal cloud structures, 30

- precipitation features, cloud temperature, and MODIS radiative measurements to
- enhance classification accuracy. These CloudSat products provide critical
- microphysical parameters and cloud classification necessary for understanding ice
- cloud properties.
- CALIPSO can monitor the vertical distribution characteristics of clouds and
- aerosols, automatically identify aerosol types, and provide global aerosol horizontal
- distribution characteristics and vertical distribution information (Zheng et al., 2018).
- Liu et al. (2008) also conducted aerosol detection using CALIPSO, further confirming
- its effective aerosol detection capabilities.
- Dust aerosols exhibit strong ice-nucleating activity and represent an important
- global source of INPs (Hoose and Möhler, 2012; Murray et al., 2012; Ladino Moreno
- 12 et al., 2013). Meanwhile, sampling studies during biomass burning conducted by
- Prenni et al. (2012) and McCluskey et al. (2014) indicate that particles from biomass
- 14 combustion constitute a significant regional source of INPs, particularly when other
- 15 effective INPs are scarce. Therefore, this study primarily focuses on the role of dust
- and smoke aerosols. This study employs information from the Level-2 Version 5
- 17 kmCLay standard products of the CALIPSO satellite data spanning from 2006 to
- 18 2016 (except 2011) to assess the impact of dust and smoke aerosols on the formation
- 19 of cirrus clouds.

## 2.2 Reanalysis Data

- To investigate atmospheric context for the satellite observations, this study
- 22 utilizes ERA5 reanalysis data from the European Centre for Medium-Range Weather
- 23 Forecasts (ECMWF). ERA5 provides hourly global data at a spatial resolution of
- $0.25^{\circ} \times 0.25^{\circ}$  across 37 vertical pressure levels, covering the period from 1979 to
- the present (Xie et al., 2021). The key variables used in this study are specific
- humidity and vertical wind velocity, which are essential for analyzing atmospheric
- conditions related to cirrus cloud formation and deep convective vertical transport.

## 28 2.3 Research Methods

- The focus area of this study is the TP and its surrounding regions, spanning from
- 30 66°E to 106°E and 24°N to 40°N. The original orbital data of the DARDAR-Nice

1 PRO product, 2B-CLDCLASS-LIDAR classification product and 2B-CWC-RO cloud

product are interpolated into grid point data with a resolution of  $2^{\circ} \times 2^{\circ}$  based on the

3 method outlined in Wang et al. (2023). The 2B-CLDCLASS-LIDAR deep convective

cloud product was used to quantify deep convection. For each grid cell and time

interval, the presence of one or more deep convective clouds was counted as a single

event, irrespective of the number of profiles exhibiting convection. These events were

then summed over all intervals to yield the total number of deep convection

occurrences per grid cell. For the investigation of N<sub>i</sub>, statistical analysis was

conducted at intervals of 60 m, based on the vertical resolution of the DARDAR-Nice

PRO product. Data points with large uncertainties were set to NaN to minimize bias

in the statistics.

10

12

13

14

15 16

18

19 20

21

22

23

24

2526

The horizontal distribution of  $N_i$  may be influenced by uneven sample distribution resulting from the varying occurrence frequency of ice particles across different layers. To address this, we normalized  $N_i$  for each grid during the ten summer seasons over the TP, to obtain the horizontal distribution of  $N_i$  for cirrus clouds. The normalization process is presented in Eq. (1):

$$y = \frac{\sum_{i=1}^{n} x_i}{n} \tag{1}$$

Where  $x_i$  is the sum of  $N_i$  where the temperature is below -30°C. n is the number of occurrences in the corresponding gird during the study period. y is the normalized  $N_i$  in the corresponding gird.

To compute the vertical distribution of  $N_i$ , each profile is analyzed layer by layer. For each profile, if  $N_i$  is present in any layer, the profile is counted as 1; this count is used for normalizing the total number of profiles. For each layer, the  $N_i$  from all profiles are summed and then divided by the total number of counted profiles, yielding the normalized  $N_i$  for that layer. The detailed calculation method is given in Eq. (2):

$$P_{j} = \frac{\sum_{i=1}^{N_{matal}} x_{i,j}}{\sum_{i=1}^{N_{total}} C_{i}}$$
(2)

where  $P_i$  represents the normalized  $N_i$  for layer j,  $N_{total}$  is the total number of

profiles included in the analysis,  $x_{i,j}$  is the  $N_i$  in layer j of profile i,  $C_i$  is the profile count ( $C_i = 1$  if  $N_i$  is present in at least one layer of profile i, and  $C_i = 0$  otherwise). In the absence of INPs in the atmosphere, ice crystal formation occurs primarily through the homogeneous nucleation of water vapor. It is generally acknowledged that -38°C represents the threshold for homogeneous ice nucleation of water vapor (Koop and Murray, 2016). Traditionally, the identification of homogeneous nucleation has relied primarily on temperature thresholds. However, due to the continuous dynamic growth of ice particles through condensation, accurate simulation remains challenging. Moreover, purely homogeneous nucleation events are extremely rare in the natural atmosphere.

A novel approach is proposed to identify homogeneous nucleation by leveraging aerosol classification data from the CALIPSO satellite over the TP during the summer from 2006 to 2016. Specifically, when aerosol types are classified as 'clean', it indicates a low concentration of INPs, favoring the dominance of homogeneous nucleation in ice crystal formation. Kim et al. (2018) performed a statistical analysis of different aerosol types in this product and found that 'clean' aerosols account for only about 1% of occurrences, representing background aerosol with very low concentration, which further supports the validity of this assumption. Grid points identified exclusively with 'clean' aerosol conditions are therefore considered to have undergone only homogeneous nucleation. Although CALIPSO has limitations in detecting aerosols in the upper troposphere due to sparse aerosol layers and weak signal strength (Mao et al., 2022), resulting in some micro-scale undetected aerosols may introduce small deviations in individual values, this study focuses on large-scale ice crystal formation events over the TP during the ten summer seasons from 2006 to 2016 (except 2011), including both homogeneous and heterogeneous nucleation.

Therfore, the overall comparison, statistical results, and main conclusions remain

robust.

7

17

2122

23

### 3 Results and Discussion

### 3.1 Distribution characteristics of $N_i$ over the TP

Based on the DARDAR-Nice PRO product, this study analyzes the spatial variation of  $N_i$  across all layers where the temperature is below -30°C during the study period. The horizontal distribution of N<sub>i</sub> in Fig. 1 demonstrates that the average concentration over the TP is 187.48 L-1 during the study period. This value is comparable to the annual mean reported by Gryspeerdt et al. (2018), who used DARDAR-Nice data from 2006 to 2013 to study global  $N_i$  and found values of around 150 L<sup>-1</sup> over the TP. Furthermore, it falls within the range obtained from in-situ measurements of high-altitude summer cirrus clouds reported by Cairo et al. (2023), who observed N<sub>i</sub> ranging from 0.1 to 10<sup>4</sup> L<sup>-1</sup> over the Himalayas during July and August 2017, coinciding with the ASM. The agreement with the independent in-situ observations reinforces the reliability of our satellite-derived  $N_i$  estimates and indicates that the DARDAR-Nice product can effectively capture the microphysical properties of cirrus clouds in such extreme environments. The average concentration in the south (24-30°N, 66-106°E) is significantly higher than other areas, reaching 213.12 L-1, with the maximum value located in the north-central region of India (24-26°N, 78-80°E), reaching 252.95 L<sup>-1</sup>. Over the north, including the Xinjiang, Inner Mongolia, the north of the Qilian Mountains and the Kunlun Mountains,  $N_i$  is only 142.75 L<sup>-1</sup>, only two-thirds of  $N_i$  compared with that in the southern region.

Fig. 1. Horizontal distribution of the averaged N<sub>i</sub> during the summer from 2006 to 2016 (except

2 2011) over the TP. The green line is the border of the TP. The black solid lines represent the

3 standard error of  $80 L^{-1}$  and  $100 L^{-1}$ . The black dotted line represents the value of  $60 L^{-1}$ .

We investigated the relationship between the incidence of deep convective clouds, INPs, and  $N_i$  at all grid points over the TP based on the different formation mechanisms of cirrus clouds. As shown in Fig. 2a, deep convection occurrences (DCO) is significantly higher in the southern and southeastern regions of the Plateau, where the  $N_i$  tend to be elevated (Fig. 1). Also,  $N_i$  revealed a positive nonlinear relationship with DCO, with a coefficient of determination (R<sup>2</sup>) of 0.55 (p<0.01) and a root mean square error (RMSE) of 24.03 L<sup>-1</sup>, indicating that deep convective activity plays a significant role in modulating  $N_i$  over the TP. During the ASM, frequent deep convection in the southern TP facilitates the transport of warm, moist air from the Indian Ocean and the Bay of Bengal to higher altitudes. This process enhances homogeneous nucleation and ice crystal formation, thereby increasing  $N_i$  over the southern region.

In addition to convective activity, the presence of INPs also plays a critical role in modulating  $N_i$  over the TP. Zhao et al. (2018), using nine years of satellite observations, demonstrated that ice crystal formation is regulated not only by the availability of INPs but also by ambient water vapor conditions. This highlights the crucial role of moisture in modulating cirrus formation pathways and underscores the need to account for water vapor when assessing the influence of INPs. Therefore, when investigating the relationship between INPs and  $N_i$ , directly comparing INPs and  $N_i$  across all grid cells may lead to misleading or inaccurate conclusions, as it cannot accurately capture their relationship due to differing atmospheric conditions, such as water vapor content, in each grid cell. For example, high  $N_i$  in one grid cell could be due to abundant water vapor rather than the effect of INPs, while low  $N_i$  in another grid cell could mask the influence of INPs. If the analysis is restricted to grid cells with similar atmospheric conditions, direct comparison is feasible. However, the TP exhibits large spatial variations in atmospheric properties between the north and

south. To account for this, the IWC confined INPs concentration (ICIC) is calculated

2 by the logarithm of the ratio between the number of smoke (dust) particle occurrences

3 and the IWC in each grid cell. This approach effectively normalizes the atmospheric

conditions of each grid cell, allowing a more accurate assessment of the impact of

5 INPs on  $N_i$ .

Fig.3a shows the spatial distribution of this metric, revealing that ICIC is predominantly concentrated in the northern part of the Plateau, with significantly lower values in the south. Moreover, an inverse nonlinear relationship is observed between ICIC and  $N_i$ , with a coefficient of determination (R<sup>2</sup>) of 0.61 (p<0.01) and a root mean square error (RMSE) of 21.94 L<sup>-1</sup>, indicates that the quantity of ICIC has a significant impact on the  $N_i$  over the TP. While the  $N_i$  mainly arises from homogenization nucleation, heterogeneous nucleation of INPs promotes the formation of larger ice crystals (DeMott et al., 2010; Khvorostyanov et al., 2006) by absorbing a large amount of water vapor and destroying the conditions for homogeneous nucleation. The inhibitory effect of heterogeneous nucleation on homogeneous nucleation becomes more pronounced with an increase in INPs content, leading to a lower  $N_i$ . These two factors, namely the increased convective cloud frequency in the south and the elevated INPs levels in the north, are the primary contributors to the observed spatial pattern of  $N_i$ , which tends to be higher in the south and lower in the north.

Fig. 2. (a) Horizontal distribution of DCO and (b) the relationship with  $N_i$  during the summer from 2006 to 2016 (except 2011) over the TP.

Fig. 3. (a) Horizontal distribution of the ICIC and (b) the relationship with  $N_i$  during the summer

3 from 2006 to 2016 (except 2011) over the TP.

# 3.2 Generation mechanism of ice crystal formation

# 3.2.1 Contribution of the homogeneous nucleation

Due to the condensation growth of cirrus cloud ice crystals in the upper atmosphere after nucleation, the observed ice crystal particle size in the satellite observation dataset only represents the post-growth effect, rendering it impossible to distinguish the contribution of different nucleation mechanisms to ice crystal size. Thus, this study considered the contribution of different nucleation mechanisms to the formation of cirrus cloud ice crystals by examining changes in the  $N_i$ .

Fig. 4 depicts the vertical distribution of the  $N_i$  from satellite observations and homogeneous nucleation. The satellite observations indicate that the  $N_i$  initially slowly increases with height and reaches its maximum of 67.56 L<sup>-1</sup> at 14 km, and follows a decreasing trend with height up to 19 km. The vertical variation of  $N_i$  from homogeneous nucleation and observation both show an overall 'V' shaped distribution. However,  $N_i$  derived from homogeneous nucleation is consistently higher than the satellite observations at corresponding altitudes. Specifically, the number concentration from homogeneous nucleation peaks at 14 km with a value of 94.35 L<sup>-1</sup>, which coincides with the altitude of the observed peak.

It is widely accepted that the formation of larger ice crystals through heterogeneous nucleation processes takes precedence over homogeneous nucleation (Shi et al., 2017; Barahona and Nenes, 2009). In fact, heterogeneous nucleation is the dominant ice formation mechanism at temperatures above -38 °C, whereas

19 20

21

homogeneous nucleation occurs only when the temperature drops below -38 °C. 1 Although homogeneous nucleation is the major contributor to the  $N_i$  (Cantrell and 2 Heymsfield, 2005), heterogeneous nucleation has relatively low formation 3 4 requirements and often occurs prior to homogeneous nucleation. During this process, 5 a large amount of water vapor is absorbed, which further suppresses the formation of ice crystals through homogeneous nucleation. This mechanism also explains why the 6 7 observed  $N_i$  is significantly lower than that produced only by homogeneous nucleation. Additionally, it is also worth noting that the observed  $N_i$  slightly exceeds the values 8 from homogeneous nucleation below approximately 12 km. This is likely because 9 10 homogeneous nucleation has not yet become dominant in this layer, while the 11 observed  $N_i$  reflects prior heterogeneous nucleation events that produced a relatively 12 large number of ice crystals. Once homogeneous nucleation becomes active with 13 decreasing temperature, it rapidly generates a substantially higher  $N_i$  than observed. From a trend perspective, the 'V' shape vertical distribution and the peak position of 14  $N_i$  is determined by the role of homogeneous nucleation, while the specific values at 15 different altitudes are determined by the combined effect of homogeneous nucleation 16 17 and heterogeneous nucleation.

Fig. 4. Vertical profiles of observed and homogeneously nucleated  $N_i$ , with light shading indicating the standard error range.

To further investigate the vertical distribution characteristics of  $N_i$ , this study analyzes its spatial distribution across different latitudes and longitudes based on Fig. 5. In the zonal cross-section (Fig. 5a), the  $N_i$  exhibits a pronounced maximum near 14 km altitude between 28°N and 33°N, exceeding 120 L<sup>-1</sup>. Additionally, in the meridional cross-section (Fig. 5b), a peak  $N_i$  of over 90 L<sup>-1</sup> is observed near 90°E, also centered at 14 km altitude.

Together, these zonal and meridional distributions reveal a consistent vertical structure, with peak  $N_i$  occurring near 14 km, primarily driven by homogeneous nucleation processes that dominate at these altitudes(Fig. 4). In contrast,  $N_i$  exhibits significant variability across both latitudinal and longitudinal directions, which is likely influenced by the spatial heterogeneity of aerosol-derived INPs and the localized intensity of deep convection.

Fig. 5. (a) The zonal distribution of  $N_i$  from 86 to 102°E for each latitude and (b) the meridional distribution of  $N_i$  from 24 to 40°N for each longitude.

## 3.2.2 The effect of deep convective activity

In addition to homogeneous nucleation, deep convective cloud anvils are another significant source of ice crystal formation in the atmosphere. Fig. 6a compares the altitude-averaged  $N_i$  under different deep convective cloud conditions, based on all

2

7

17

2122

grid points across the TP where the incidence of deep convection exceeds 5% (Fig. 2a). These selected regions represent areas with relatively frequent convective activity. The top of cirrus clouds can develop near 18 km with a relatively low  $N_i$  for the case of non-deep convection activity, and at 14 km, reaches its peak of 103.94 L<sup>-1</sup>. When deep convection activity occurs, the  $N_i$  at the same altitude is significantly higher, and at 14 km, reaches its peak of 161.84 L<sup>-1</sup>. In summer, strong upward motion over the southern TP transports a large amount of water vapor from the lower atmosphere to the upper atmosphere (Fig. 6b), resulting in a significant increase of  $N_i$  by the homogeneous nucleation process. It is evident that the N<sub>i</sub> of cirrus clouds peaks at 14 km, which is due to deep convective activity. At 14 km ( about 140 hPa), where the vertical wind speed is nearly zero,  $N_i$  accumulates significantly above this level, while a substantial amount is transported upward from below 14 km. This upward transport, combined with the accumulation, results in the peak at 14 km (Fig. 4b). Satellite observations also indicate that during the development of deep convective clouds, approximately 95% of the cloud tops are located at or below 16 km. This vertical distribution suggests that the influence of deep convection is mainly confined below 16 km. Consequently, N<sub>i</sub> above this level remains relatively unchanged, indicating limited impact from convective processes at higher altitudes.

Fig. 6. (a) Vertical profile of the  $N_i$  affected by DCO and (b) the zonal distribution of vertical winds averaged from 86 to 102°E for each latitude. The contour is specific humidity (kg kg<sup>-1</sup>).

3

## 3.2.3 Heterogeneous nucleation effect of INPs

4 aerosol content is high (Fig. 7a). The increase in  $N_i$  is primarily attributed to heterogeneous nucleation induced by INPs. Considering the frequent dust activity in 5 this region, we selected grid points with ICIC(dust) greater than -5 as the primary 6 7 study area. These grid points are predominantly located in the northern Plateau, adjacent to Xinjiang, a typical semi-arid region characterized by abundant dust 8 aerosols. These dust particles facilitate water vapor adsorption in the lower 9 atmosphere and promote ice crystal formation through heterogeneous nucleation 10 (Huang et al., 2021). 11 12 Fig. 7b illustrates the effect of dust aerosol particles on  $N_i$  in this area. The results indicate that the presence of dust significantly reduces  $N_i$  in cirrus clouds, with 13 concentrations above 12 km nearly approaching zero. During non-dust periods, 14 although INPs remain present, their activation efficiency is high, nearly all aerosol ice 15 nuclei are activated to form ice crystals, resulting in relatively weak suppression 16 17 among heterogeneous nucleation processes. In contrast, elevated dust concentrations 18 in the lower atmosphere enhance heterogeneous nucleation, thereby consuming 19 available water vapor and inhibiting additional ice crystal formation, which leads to a 20 significant reduction of  $N_i$ . This explains why  $N_i$  during non-dust periods is higher than that during dust. However, due to limited water vapor in this region, most 21 22 moisture is depleted in the lower atmosphere, causing  $N_i$  above 12 km to be nearly 23 zero. Consequently, the suppressive effect of heterogeneous nucleation inhibits ice crystal formation through homogeneous nucleation at higher altitudes, making cirrus 24 cloud development more difficult in these upper layers. Therefore, it can be concluded 25 that in regions with low water vapor content, INPs plays a dominant role in 26 27 determining  $N_i$ .

In the northern part of the TP, convective activity is relatively weak, but dust

Fig. 7. (a) Horizontal distribution of ICIC(dust) and (b) the vertical profile of the  $N_i$  affected by dust aerosols.

Besides dust aerosol, smoke aerosol particles generated by human activities are another important source of heterogeneous nucleation for cirrus clouds over the TP. In this research, grid points with ICIC(smoke) greater than -6.5 were selected as the primary research region to analyze the impact of smoke INPs on  $N_i$  (Fig. 8a).

It is observed that the presence of smoke aerosols leads to a decrease in  $N_i$ , with the maximum development height of cirrus clouds limited to around 14 km (Fig. 8b). During smoke events, ice crystal formation in the lower atmosphere is primarily governed by smoke-derived INPs, where heterogeneous nucleation dominates. The high abundance of smoke INPs causes strong competition among ice particles, suppressing additional ice crystal formation and resulting in a lower  $N_i$  compared to non-smoke events. During smoke events, the utilization efficiency of smoke INPs is nearly 100%, meaning that almost all available INPs are activated to form ice crystals through heterogeneous nucleation. Any excess water vapor not consumed by this process may still contribute to ice formation via homogeneous nucleation. This explains why  $N_i$  is generally higher during non-smoke events that fewer INPs lead to weaker suppression effects and more efficient homogeneous nucleation. However, due to the inherently low water vapor content in this region, the vertical development of cirrus clouds is constrained. Even during non-smoke events, the maximum height of cirrus clouds is limited to approximately 17 km.

Notably, homogeneous nucleation typically leads to a peak in  $N_i$  at 14 km. However, in cases involving dust and smoke aerosols, this peak is not observed. This

- is primarily because dust and smoke are concentrated in the northern TP, where the
- vertical wind speed at around 400 hPa is nearly zero (Fig. 6b). The weak vertical
- transport prevents ice crystals from being carried upward efficiently, resulting in their
- accumulation in the upper atmosphere. Therefore, the  $N_i$  peak appears near the
- homogeneous nucleation threshold temperature (-38°C), suppressing its development
- at around 14 km altitude.

Fig. 8. (a) Horizontal distribution of ICIC(smoke) and (b) the vertical profile of the  $N_i$  affected by smoke aerosols.

## **4 Conclusion**

7 8

17

19

This study analyzed the distribution characteristics and formation mechanism of cirrus cloud ice crystals during the summer of 2006 to 2016 (except 2011) over the TP, mainly using DARDAR-Nice data combined with aerosol product data.

The main conclusions are summarized as follows: (1) The  $N_i$  show clear spatial differences across the TP. The  $N_i$  is significantly higher over the southern TP, where frequent deep convection promotes ice crystal formation, compared with the northern TP, where more abundant aerosols are present and  $N_i$  remains relatively low.

- (2) Homogeneous nucleation leads to a characteristic 'V' shaped distribution of  $N_i$ . The values of  $N_i$  at varying altitudes are determined by the combined effects of both homogeneous and heterogeneous nucleation.
- (3) Deep convection activities are found to promote the concentration of ice crystals. The deep convective cloud anvil enhances the  $N_i$  at the same altitude compared to non-deep convective activities.
  - (4) The presence of dust and smoke aerosols leads to ice crystal formation via

- heterogeneous nucleation in the lower atmosphere, occurring before homogeneous
- nucleation. Heterogeneous nucleation suppresses further ice crystal formation,
- resulting in lower concentrations compared to non-dust and non-smoke conditions.
- Additionally, the vertical velocity at high altitudes drops to zero earlier, causing the
- ice crystal number concentration peak to appear prematurely.

Acknowledgements. This study was supported by the National Natural Science Foundation of China

- (NSFC, Grant Numbers: 42330603), the Open Research of Key Laboratory of Intelligent
- Meteorological Observation Technology in China Meteorological Administration (ZNGC2024ZD02),
- the Science and Technology Planning Program of Xinjiang (2022E01047), National Natural Science
- Foundation of China (42030612 and 42175179), and the Natural Science Foundation of Shanghai
- (22ZR1404000). The authors gratefully acknowledge the ECMWF for providing ERA5 data, and the
- NASA for providing CloudSat and CALIPSO data. In addition, the DARDAR-Nice product used in
- this study was obtained from the AERIS/ICARE data center.

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
