# Peer review of "A satellite observation-based analysis of the"

_EGUsphere, 2025_

## Author Comment (AC1)

Response to Referee Comments on the Study of **"A satellite observation-based analysis of the distribution and formation mechanism of ice crystal number concentration over the Tibetan Plateau"**

We sincerely thank the referee for their valuable comments and suggestions, which have greatly helped improve the quality of this manuscript. We have carefully addressed each comment and revised the manuscript accordingly, as detailed below:

**Comments from Referee 1**

I think the manuscript can be published as an ACP discussion version.
The N-i numbers in the figures make sense, the findings (retrieval products) are based on observed data, uncertainty margins are included in several figures, and the amount of speculative discussions is widely reduced.
There is no objective reason left to reject the manuscript.
However, many questions and aspects are left that have to be improved, but that can be done within the regular revision process. However, some points may be quickly improved before the discussion round will start.
Here I provide a list with these open points. The authors may decide what they can consider before the manuscript will be published as discussion version, and what they want to leave open for later (regular revision).

1、The title should be improved! Maybe: A satellite observation-based analysis of cirrus ice crystal number concentrations and underlying cirrus formation mechanisms
**Respone:We appreciate the reviewer's constructive suggestion regarding the title. Following the recommendation, the title has been revised.**

2、A table would be good to have an overview of all data sources and data products.
**Respone:We appreciate the reviewer's helpful suggestion. Following the recommendation, a new table has been added to the revised manuscript to provide an overview of all data sources and data products used in this study.**

| Source | Dataset | Variable | Duration |
|--------|---------|----------|----------|
| DARDAR | DARDAR-Nice PRO | icnc_5um | |
| | | icnc_5um_error | 2006    - |

| | | temperature | 2016 |
| --- | --- | --- | --- |
| *Cloudsat* | *2B-CWC-RO* | *ice water content* | *(except* |
| | *2B-CLDCLASS-LIDAR* | *Cloud type(deep convection)* | *2011)* |
| *Calipso* | *Level-2 Version 5 kmCLay* | *Aerosol type(dust、smoke、clean)* | |
| *ERA5* | *Reanalysis* | *specific humidity*
 *vertical wind velocity* | |

3、It would be nice to have a sketch with10-20 grid cells (3-D cubes, 2 degrees times 2 degrees times 60m vertical), defined by latidude, longitude, and height, and then to have list of the observation-based information for each of these grid cells.

**Respone:We thank the reviewer for this valuable suggestion. Following your advice, a schematic diagram has been provided below to illustrate the main computational framework used in this study.**

[Figure]

4、In this context: It took me a while to understand that you use the CALIPSO aerosol typing information (such as smoke, dust, clean) in each individual grid cell to decide, for example, that CLEAN indicates homogeneous freezing! If the grid cell is assigned to DUST, you conclude that heterogeneous ice nucleation on dust particles was responsible for the DARDAR-derived ice crystal number concentration. I hope, my interpretation is correct. However, you cannot have

aerosol and ice crystal information for a given grid cell at the same time, i.e., simultaneously. Aerosol information is only available during cloud-free conditions. So, how did you link the CALIPSO aerosol type information with the DARDAR ice number concentrations, observed at different times? A more detailed explanation is needed here. This temporal aspect could also be visualized in such a sketch.

**Respone:We thank the reviewer for this important comment. As correctly noted, CALIPSO aerosol-type information is only available during cloud-free conditions, whereas DARDAR provides Ni only when clouds are present. Therefore, the two quantities are not obtained simultaneously. To address this, we process both datasets into daily gridded fields. Because the ground tracks of CALIPSO and DARDAR are not identical, a given daily grid cell may contain cloudy observations from DARDAR (providing Ni) as well as cloud-free observations from CALIPSO (providing aerosol type). The aerosol type assigned to a grid cell thus represents the dominant aerosol environment observed under cloud-free conditions on that day, and this environmental classification is used to interpret the Ni values that occur during cloudy periods within the same grid cell. When counting the contribution of a specific aerosol type—such as dust—we include only those grid cells and days on which only dust was detected and no other aerosol types occurred, and we use the corresponding Ni values from those same grid cells. In other words, the aerosol – Ni relationship is established through daily spatial collocation rather than simultaneous measurements. For clarity, we provide a simplified schematic illustration of this temporal-spatial matching process below.**

[Figure]

5、You did not use any parameterization to compute N-i in the case of homogeneous freezing? Only the DARDAR data of N-i and the aerosol type information (CLEAN) from CALIOP observations are used? Please explain that more clearly!

**Respone: We appreciate the reviewer's question and the opportunity to clarify this point. In this study, no parameterization was applied to compute the ice crystal number concentration for homogeneous freezing. After reviewing a large body of relevant literature, we found that no existing parameterization can reliably compute Ni using only the variables available from satellite observations and reanalysis datasets.**

**Instead, Ni values were directly obtained from the DARDAR product, which provides observationally constrained estimates derived from the synergy of CloudSat radar and CALIPSO lidar measurements.The CALIPSO aerosol type information (e.g., CLEAN, DUST, SMOKE) was used only to categorize the atmospheric environment rather than to calculate Ni. Specifically, for the "CLEAN" condition, we considered grid cells and days where CALIPSO detected only**

clean air (i.e., without dust or smoke types) to represent environments dominated by homogeneous nucleation. The corresponding daily mean Ni from DARDAR was then used to characterize the ice crystal number concentration under these aerosol-free conditions.

This approach allows us to investigate the statistical relationship between aerosol environments and observed Ni, without imposing any assumptions from ice nucleation parameterizations.

6、An equation for ICIC, i.e., ICIC-type = log(type event/IWC) with type = dust or smoke, in an equation environment, not just described in the main text, would better highlight ICIC.

**Respone:We appreciate the reviewer's helpful suggestion. Following your advice, the equation defining ICIC has been explicitly added in the manuscript to better highlight this parameter.**

$$ICIC(type)= log(\frac{type_{event}}{IWC})\qquad\qquad (3)$$

*Where type is dust or smoke, $type_{event}$ is the number of events of that aerosol type, and ICIC(type) is the ICIC value corresponding to that aerosol type in the corresponding gird.*

7、Page 6, line 25: You write that the model key variables include specific humidity and vertical wind velocity. I think also the temperature belongs to the key variables!

**Respone:We thank the reviewer for this helpful suggestion. We have revised the text to explicitly include temperature as one of the key variables in our analysis. In addition, we have added the corresponding temperature profiles in Figure 6a .**

*The key variables used in this study are specific humidity and vertical wind velocity from ERA5, as well as temperature from satellite observations. Together, these three variables are essential for analyzing atmospheric conditions related to cirrus cloud formation and deep convective vertical transport.*

8、I did not understand, what the dashed (60 L-1) and solid lines (80, 100 L-1) in Fig. 1 show. Is the dashed line a mean value of a certain grid cell ensemble? Please explain!

**Respone:We appreciate the reviewer's comment. More detailed explanations have been added in the figure caption to clarify the meaning of the dashed and solid contour lines. Specifically, the black dashed line (60 L$^{-1}$) and solid lines (80 L$^{-1}$) represent the isolines derived from the mean standard error of Ni calculated at each grid.**

*Fig. 1. Horizontal distribution of the averaged Ni during the summer from 2006 to 2016 (except 2011) over the TP. The green line is the border of the TP. The black solid lines represent the standard error of 80 L-1 and 100 L-1. The black dotted line represents the value of 60 L-1. These contours were obtained through calculating the standard error of the averaged Ni at each grid.*

9、In Figures 2 and 3, we have the regression equations with y= ax +b. Is this y the same as the y

in Eq.(1)? Please clarify!

**Respone:We appreciate the reviewer's careful observation. Yes, the variable y in the regression equations shown in Figures 2 and 3 is the same as y in Eq. (1). For clarity, we have revised the variable names in the equations.**

10、Is Pj (Eq.2) used somewhere? If not, Eq.(2) can be removed. Maybe I overlooked the usage.

**Respone:We appreciate the reviewer's careful reading. The variable Pj in Eq. (2) is indeed used in the manuscript, specifically in Section 2.3, to calculate the vertical distribution of ice crystal number concentration within each grid.**

*To compute the vertical distribution of Ni, each profile is analyzed layer by layer.*

11、In Figures 7b and 8b two curves (green and red) are shown, but the figure caption text for 7b and for 8b only explains the red curves.

**Respone:We thank the reviewer for pointing this out. The figure captions for Figures 7b and 8b have been updated to include descriptions of both the red and green curves.**

*Fig. 7. (a) Horizontal distribution of ICIC(dust) and (b) the vertical profile of the Ni affected by dust and non-dust events.*

12、In both figures 7b and 8b, we have the non-dust and non-smoke Ni profiles. Why are these two green curves different? I was spontaneously thinking that the same data set is used for these non-dust and non-smoke scenarios, but this is obviously not the case. Please expand the discussion on this! Why are the green curves in Figs. 7b and 8b different? What did I miss? Obviously, different data sets with different tropopause distributions are used.

**Respone:We thank the reviewer for this insightful comment. The green curves in Figures 7b and 8b appear different because they are not derived from the same set of grid cells. Specifically, the non-dust Ni profile is calculated from grid cells and time periods where dust events did not occur, while the non-smoke Ni profile is calculated from grid cells and time periods where smoke events did not occur. We specifically focus on regions with strong dust or smoke events because including the entire Tibetan Plateau, which contains areas with weak or no aerosol activity, could amplify other confounding effects and potentially lead to less accurate conclusions. By selecting grid cells with strong aerosol occurrences, our analysis more accurately isolates the influence of dust and smoke on Ni, providing more robust and reliable results.**

13、To my opinion, if you show dust-related and smoke-related cirrus number concentrations up to 12 km and up to 16 km, respectively, one needs to present the corresponding profiles with aerosol type information as well. I mean, for example, the number of dust events as a function of height. Dust is usually transported at heights below 6 km, and not in the height range from 8-12 km! Was there really dust up to 12 km? How many dust events are considered in the dust N-i profiles in Fig. 7b? Smoke, on the other hand, can easily reach the tropopause (in the case of strong cloud convection and thunderstorms). Please clarify this point!

**Respone:We thank the reviewer for this important comment. More detailed explanations have been added in Section 2.3.**

*Although CALIPSO provides detailed vertical profiles of aerosols, this study does not explicitly use*

*the height-resolved information. Instead, the aerosol occurrence is analyzed at the grid-cell level without distinguishing altitude. This approach is adopted for two main reasons. First, CALIPSO's aerosol detection is most reliable in the lower troposphere, while its sensitivity decreases significantly at higher altitudes due to signal attenuation and the difficulty of distinguishing aerosols from thin cirrus clouds (Mao et al., 2022). Therefore, focusing on overall aerosol occurrence within each grid ensures better data consistency and avoids potential misclassification errors. Second, the Ni analyzed in this study corresponds to temperatures below -30 ℃, the relevant aerosols are those that can influence cloud formation through vertical transport or large-scale dynamical processes, rather than being co-located at the same altitude. Hence, by integrating aerosol occurrence over the entire column within the same grid, the analysis effectively captures the overall influence of low-level dust or smoke aerosols on upper-tropospheric ice clouds, without introducing additional uncertainty from vertical matching. Therfore, the overall comparison, statistical results, and main conclusions remain robust.*

14、Page 16, line 22, and page 19, line1: please provide the precise height range, when you mention … the lower atmosphere.
**Respone:We thank the reviewer for pointing this out. The manuscript has been updated so that at the first mention of the lower atmosphere, the precise height range (below 12 km) is specified. Subsequent mentions retain the term "lower atmosphere" without repeating the height range for readability.**

**Comments from Referee2**

This manuscript is a revised version of one I previously reviewed. I can see that the authors have implemented several improvements based on earlier comments made during the quick review round. However, not all concerns have been adequately addressed.
Overall, I still believe the paper has value in presenting the state of ice clouds, particularly in terms of ice crystal number concentration (Ni), over the Tibetan Plateau region. This is a scientifically relevant topic, and the authors are generally using appropriate datasets and a reasonable methodological approach. That said, the manuscript remains somewhat unfocused.

1、Many sentences throughout the manuscript are confusing or imprecise in describing the physical processes being studied. For example, statements such as "Cirrus clouds are located at the upper middle-lower troposphere", "It is supposed that homogeneous nucleation is a dominant mechanism to decide Ni", "Cirrus ice crystal particle is at the mercy of three mainstream mechanisms: deep convective cloud anvil overflow, homogeneous nucleation and heterogeneous nucleation", and "Ice crystal formation occurs primarily through the homogeneous nucleation of water vapor" (to quote a few) are unclear or misleading.
**Respone:Based on the reviewer's comments, we have carefully revised the relevant textual descriptions in the manuscript to improve clarity and accuracy.**

2、The instrumentation section is also a bit confusing. The use of several instruments is justified

by the intent to rely on well-validated data, which is reasonable. However, the instruments involved have different sensitivities and detection capabilities. For example, DARDAR combines lidar and radar, 2D-CWC-RO uses radar only, and 2D-CLOUDCLASS-LIDAR uses lidar only. This raises concerns about whether the different instruments are observing comparable cloud scenes across the various analyses presented. This concern should be discussed by the authors. Moreover, while DARDAR was excluded due to concerns over IWC validation, CALIPSO aerosol data were still used despite acknowledged limitations in aerosol detection.The manuscript claims that these limitations are negligible due to long-term averaging, but this assumption should also be discussed and justified more carefully.

**Respone:We thank the reviewer for this valuable comment. We fully acknowledge the differences in sensitivity and detection capabilities among the instruments used in this study. Similar concerns were raised by reviewers in previous rounds of evaluation, who suggested that we should rely as much as possible on datasets and related variables that have been thoroughly evaluated and validated. Following their suggestions, we adopted this approach in the present study to minimize inconsistencies among different instruments and to ensure the comparability and reliability of our results.**

**Regarding the CALIPSO aerosol data, we agree that its aerosol detection capability has inherent limitations. Additional clarification have been added in the revised manuscript in Section 2.1 and 3.2.**

*In this study, both daytime and nighttime satellite observations are included, the aerosol information is used to characterize climatological, grid-cell-averaged aerosol occurrence rather than instantaneous cloud-aerosol collocation.*

*Although CALIPSO provides detailed vertical profiles of aerosols, this study does not explicitly use the height-resolved information. Instead, the aerosol occurrence is analyzed at the grid-cell level without distinguishing altitude. This approach is adopted for two main reasons. First, CALIPSO's aerosol detection is most reliable in the lower troposphere, while its sensitivity decreases significantly at higher altitudes due to signal attenuation and the difficulty of distinguishing aerosols from thin cirrus clouds (Mao et al., 2022). Therefore, focusing on overall aerosol occurrence within each grid ensures better data consistency and avoids potential misclassification errors. Second, the Ni analyzed in this study corresponds to temperatures below -30 ℃, the relevant aerosols are those that can influence cloud formation through vertical transport or large-scale dynamical processes, rather than being co-located at the same altitude. Hence, by integrating aerosol occurrence over the entire column within the same grid, the analysis effectively captures the overall influence of low-level dust or smoke aerosols on upper-tropospheric ice clouds, without introducing additional uncertainty from vertical matching. Therfore, the overall comparison, statistical results, and main conclusions remain robust.*

3、The definition of quantities also remains unclear. The "Nice" metric is unconventional (it represents the horizontal and vertical average of all Ni values within each 2° × 2° grid cell at temperatures below −30 ° C) but understandable. However, the reported mean value of around 190 L$^{-1}$ seems inconsistent with Figure 4, even when considering standard deviations, I am not sure to understand why.

**Respone:We thank the reviewer for this comment. The apparent discrepancy arises because**

the two calculations of Ni are based on different methods. To clarify this distinction, we have provided a schematic diagram below illustrating the main computational framework. We believe that this visual aid will help readers better understand the differences between the two calculation approaches.

[Figure]

4、 This suggests that the mean might be biased by outliers, making it difficult to interpret the absolute value, though the trends are likely still meaningful. The comparison with annual means from Gryspeerdt et al. (2018) is also problematic, since that study used values only near cloud top, and thus the datasets are not directly comparable. This shows difficulties to understand this novel metric, which needs to be better explained to the reader to fully understand the work in this manuscript.

**Respone: We thank the reviewer for raising these concerns. We have revised the relevant description in Section 3.1.**

*This value is higher than the approximately 150 L-1 over the TP reported by Gryspeerdt et al. (2018), who used DARDAR-Nice data from 2006 to 2013 to study global Ni but focused only on cloud-top statistics. Considering that our analysis includes all layers below -30 ° C, the higher Ni is reasonable and consistent with physical expectations, which also indirectly supports the reliability of our results.*

5、 The ICIC (INP concentration) parameter, defined as the logarithm of the ratio between INP number (smoke or dust) and IWC, is not well justified. It remains unclear whether the observed relationship between Ni and ICIC might simply reflect an underlying relationship between Ni and IWC, as hinted by the linear correlation shown in Figure 3.

**Respone:We thank the reviewer for this insightful comment. To verify that the relationship between Ni and ICIC reflects the influence of INPs rather than being dominated by IWC, we calculated the partial correlation coefficients between INP concentration and Ni while**

**controlling for IWC. The results indicate that INPs and Ni remain negatively correlated even when IWC is held constant, confirming that the ICIC parameter effectively captures the influence of INPs on Ni and supporting the reliability of this approach in our study(3.1).**

*In principle, restricting the analysis to grid cells with broadly similar atmospheric conditions would allow a more direct comparison. However, the TP exhibits pronounced spatial heterogeneity, especially between its northern and southern regions. To partially account for differences in moisture-related thermodynamic conditions, this study introduces the IWC confined INPs concentration (ICIC), defined as the logarithm of the ratio between the occurrence number of smoke (or dust) particles and IWC within each grid cell (Eq. 3). By standardization, this metric improves the comparability of the analysis to some extent. To further demonstrate the robustness of this normalization, we compute the partial correlation between INPs and Ni after removing the effect of IWC. The resulting coefficient, r = -0.38, confirms that the ICIC formulation effectively reduces moisture-related confounding.*

6、Also, a more major scientific concern - the paper does not discuss ice crystals of liquid origin. Are they considered irrelevant in this region? This would be surprising, as liquid-origin ice, formed through homogeneous freezing of water droplets, should play a significant role in deep convective cases. Yet, the manuscript only mentions homogeneous formation without distinguishing between liquid-origin and in situ formation. I had raised this point in my previous review, but it has not been directly addressed (but shortly answered, nonetheless).

**Respone:We thank the reviewer for this insightful comment. In the revised manuscript, we have clarified that homogeneous nucleation in our study includes liquid-origin ice, formed from the freezing of supercooled water droplets in Section 2.3. In addition, further explanation(3.1) has been added in the discussion of convective influences to better describe the role of deep convection in supplying liquid-origin ice and its potential impact on the observed Ni.**

*In the absence of INPs in the atmosphere, ice crystal formation occurs primarily through homogeneous nucleation. It is generally acknowledged that temperatures near $-38$ °C represent the threshold for homogeneous freezing of supercooled water droplets and aqueous aerosol particles under sufficiently high ice supersaturation (Koop and Murray, 2016; Duft and Leisner, 2004; Murray et al., 2010). Traditionally, the identification of homogeneous nucleation has relied primarily on temperature thresholds. However, due to the continuous dynamic growth of ice particles through condensation, accurate simulation remains challenging. Moreover, classical nucleation theory suggests that ice formation under purely homogeneous freezing conditions is generally considered to be uncommon in the natural atmosphere (Maeda, 2021).*

*We investigated the relationship between the incidence of deep convective clouds, INPs, and Ni at all grid points over the TP based on the different formation mechanisms of cirrus clouds. As shown in Fig. 2a, deep convection occurrences (DCO) is significantly higher in the southern and southeastern regions of the Plateau, where the Ni tend to be elevated (Fig. 1). Also, Ni revealed a positive nonlinear relationship with DCO, with a coefficient of determination ($R^2$) of 0.55 (p<0.01) and a root mean square error (RMSE) of 24 $L^{-1}$, indicating that deep convective activity plays a significant role in modulating Ni over the TP. During the ASM, frequent deep convection in the southern TP facilitates the transport of warm, moist air and water droplets from the Indian Ocean and the Bay of Bengal to higher altitudes (He et al., 2019). Under moist conditions, ascending air*

*parcels are more likely to experience a prolonged period of ice supersaturation, thereby increasing the probability of exceeding the supersaturation threshold required for homogeneous ice nucleation. In humid environments, air parcels can therefore maintain supersaturated conditions for a longer duration, making homogeneous nucleation more likely to dominate under such circumstances (Zhao et al., 2018). By contrast, heterogeneous nucleation is initiated by INPs and typically requires a lower ice supersaturation threshold, allowing it to occur earlier during ascent (DeMott et al., 2010). As a result, in environments with abundant water vapor, homogeneous nucleation may gain a relative advantage in competition with heterogeneous nucleation, favouring the formation of higher Ni.*

*This interpretation is consistent with the relatively high Ni observed over the southern TP during summer. However, within an observational framework alone, the respective contributions of dynamical conditions, aerosol properties, and thermodynamic processes cannot be fully disentangled. The interpretation presented here should therefore be regarded as a qualitative explanation based on physical consistency rather than a definitive attribution.*

**Comments from Anonymous Referee3**

The paper makes use of downloaded data sets of combined CALIOP and CloudSAT observations (ice-phase retrieval products) and observations of the aerosol type (dust, smoke, clean=background sulfate, derived from CALIOP observations) in the upper troposphere over the Tibetan Plateau. Based on these downloaded products the authors discuss, mostly in a speculative way, cirrus formation processes (heterogeneous vs homogeneous ice nucleation) and what the impact of deep and strong cumulus convection to cirrus formation and properties is.

1、It remains unclear how they obtained the CALIOP aerosol information and how the authors combined this aerosol information with the cirrus formation (especially the ICNC data sets, ICNC= ice crystal number concentration). The authors show the final results (derived from the complex downloaded data fields) only. The reader has no chance to check the quality of the basic data, and how the authors processed the data. There are no case studies (individual scenes with vertical profiles of all downloaded and used cirrus and aerosol data ) in the manuscript that would offer the chance to check the data quality and the ways they combined the different aerosol and cirrus data sets.

**Respone:We thank the reviewer for raising this important point regarding data transparency and processing. In the revised manuscript, additional information has been provided to clarify how CALIOP aerosol data were obtained. Specifically, all datasets used in the analysis are now explicitly listed and described, and the relevant processing steps and formulations applied to derive the final variables have been clarified.**

**With respect to the relationship between aerosol and cirrus datasets, further explanation has been added to describe the methodology used to combine CALIOP aerosol information with cirrus cloud observations. In addition, a schematic figure has been included in the figure below to illustrate the data processing workflow and the conceptual linkage between aerosol occurrence and cirrus cloud properties, providing a clearer overview of how the different datasets are integrated.**

[Figure]

2、In the discussion, the authors ignore the impact of dynamical conditions (gravity waves, large scale lofting, and lofting by orographic surface structures, etc.). Updrafts are needed to create the ice supersaturation conditions that are required to initiate nucleation bursts. Also, the impact of ice crystal sedimentation is mentioned only at the end of the manuscript.

The quality of the paper is very low. The discussion is filled with speculative statements. The authors must clearly indicate that they present hypothetical conclusions drawn from the observations.

**Respone:We thank the reviewer for these comments. We acknowledge that the roles of dynamical conditions, including gravity waves, large-scale ascent, orographic lifting, and ice crystal sedimentation, were not sufficiently emphasized in the original version of the manuscript. In response, substantial revisions have been made throughout the discussion to explicitly account for these dynamical influences and to clarify their role in modulating ice supersaturation and ice nucleation processes.**

**In addition, the discussion has been systematically revised to clearly indicate that the interpretations presented are hypothesis-driven and based on statistical relationships inferred from satellite observations, rather than definitive causal conclusions. The specific revisions**

**addressing these issues are detailed in the point-by-point responses below.**

In the present form, the manuscript cannot be accepted! Major revisions are need!

Point-by-point comments:

1、Page 5, line 15: To my opinion, one should include the ICNC solutions for particles with sizes > 25 micrometer as well. At least in some cases, >5 and >25 micrometer solutions should be shown and compared.

**Respone:We thank the reviewer for this suggestion. The focus of this study is to investigate the regional formation mechanisms of ice crystal number concentration (Ni) over the Tibetan Plateau, rather than the size-dependent microphysical evolution of individual ice crystals. In satellite retrievals such as DARDAR-Nice, ice particles undergo continuous condensational growth after nucleation, meaning that the retrieved particle sizes primarily reflect post-growth conditions rather than the original nucleation sizes. Consequently, distinguishing Ni based on thresholds of >5 μm and >25 μm does not directly provide physically meaningful information on the nucleation mechanism itself, as size-based thresholds mainly reflect post-growth effects.**

**Nevertheless, following the reviewer's suggestion, additional analyses using an ice crystal size threshold of >25 μm have been performed and are now discussed in the figure below. The corresponding results are shown in the newly added figure, where the spatial and vertical patterns of Ni derived using the >25 μm threshold are compared with those obtained using the >5 μm threshold. The comparison indicates that the main conclusions regarding the regional characteristics and inferred formation mechanisms of Ni remain largely unchanged. This supports the robustness of the results presented in this study and justifies the use of Ni for particles larger than 5 μm as a representative measure of total ice crystal number concentration.**

[Figure]

[Figure]

[Figure]

2、Page 6, line 6: What are the criteria to identify dust in the CALIOP measurements? What are the criteria to identify wildfire smoke in the CALIOP measurements? How do you relate (link) vertical profiles of dust-filled    or smoke-filled pixels with vertical profiles of cirrus-filled pixels? Figures of individual cirrus observations are needed. One case with cirrus developing in dust, one case with cirrus evolution in smoke, one case in clean air!

You do not have aerosol and cirrus information    in a given pixel at the same time! How do you combine aerosol and cirrus information?

**Respone:The aerosol information used in this study is obtained directly from the standard CALIOP aerosol products, without applying additional or non-standard processing. The identification of dust and wildfire smoke follows the aerosol subtype classification provided by CALIOP. With respect to the correspondence between aerosol and cirrus observations, aerosol and cirrus information are not required to be collocated at the individual pixel level. Instead, their relationship is established statistically at the grid-cell level. To clarify this procedure, a schematic figure is provided below to illustrate how aerosol occurrence and cirrus cloud properties are combined in the analysis.**

[Figure]

3、Page 6, line 13: What about recent smoke-cirrus papers: Mamouri et al., ACP, 2023, Ansmann et al.,    ACP, 2025.

**Respone:Thank you for pointing out these recent studies. The manuscript has been revised to include the relevant smoke－cirrus literature, including Mamouri et al. (ACP, 2023) and Ansmann et al. (ACP, 2025), and the discussion has been updated accordingly.**

*Dust aerosols exhibit strong ice-nucleating activity and represent an important global source of INPs (Hoose and Möhler, 2012; Murray et al., 2012; Ladino Moreno et al., 2013). Meanwhile, sampling studies during biomass burning conducted by Prenni et al. (2012) and McCluskey et al. (2014) indicate that particles from biomass combustion constitute a significant regional source of INPs, particularly when other effective INPs are scarce. In addition, recent observational analyses by Mamouri et al. (2023) and Ansmann et al. (2025) suggest that smoke aerosols can exert a substantial influence on ice crystal formation at altitudes while temperatures fall below −30 °C. Therefore, this study primarily focuses on the role of dust and smoke aerosols. This study employs information from the Level-2 Version 5 kmCLay standard products of the CALIPSO satellite data spanning from 2006 to 2016 (except 2011) to assess the impact of dust and smoke aerosols on the formation of cirrus clouds.*

4、Page 6, line 12-16: Is smoke ice-active at temperatures around -30°C?  Is there any information about ice nucleation efficiency of smoke at a function of temperature in the papers you mention here?

**Respone:Thank you for this insightful comment. Additional clarification regarding the ice-nucleating activity of smoke aerosols at temperatures below −30 °C has now been included in Section 2.1 to ensure a more complete and accurate description of the relevant observational evidence.**

*In addition, recent observational analyses by Mamouri et al. (2023) and Ansmann et al. (2025) suggest that smoke aerosols can exert a substantial influence on ice crystal formation at altitudes while temperatures fall below −30 °C.*

5、Page 7, line 19: ···. the number of occurrence of ···. what?

**Resopne: Thank you for this comment. The formula has been slightly revised, and the corresponding description has been corrected to clarify the definition of the number of occurrences.**

$$y = \frac{\sum_{i=1}^{Num} x_i}{\sum_{i=1}^{Num} m_i} \qquad (1)$$

*Where $x_i$ is the sum of Ni where the temperature is below -30 °C. Num is the total number of profiles included in the analysis. $m_i$ is the effective layers within the corresponding grid cell for which Ni is greater than 0. y is the normalized Ni in the corresponding gird.*

6、Page 8, lines 20-22: Grid points identified as 'clean' are therefore considered to have undergone homogeneous nucleation. Again the question arise: How is 'clean' defined in the CALIOP data base, what are the criteria for 'clean'? During cirrus observations, you do not have aerosol profile information, and during aerosol profile observations there is probably no cirrus information in the pixels.

**Respone:We thank the reviewer for this comment. In this study, "clean" grid points are defined based on the absence of detected aerosol occurrence in the CALIOP aerosol subtype product at the grid-cell level, following the standard CALIOP aerosol classification. The aerosol**

information is used to characterize climatological aerosol conditions and grid-cell－averaged aerosol occurrence, rather than instantaneous cloud－aerosol collocation at the pixel level (2.1).

As noted by the reviewer, aerosol and cirrus information are generally not available simultaneously within the same CALIOP pixel due to lidar attenuation by optically thick cirrus. Therefore, the identification of "clean" conditions does not rely on vertical aerosol profiles colocated with individual cirrus observations. Instead, "clean" grid points are interpreted as regions with persistently low aerosol occurrence, where ice formation is more likely to occur under conditions favorable for homogeneous freezing.

Additional clarification of this definition and the associated limitations has been added in Section 3.2 of the revised manuscript.

*In this study, both daytime and nighttime satellite observations are included, the aerosol information is used to characterize climatological, grid-cell-averaged aerosol occurrence rather than instantaneous cloud-aerosol collocation.*

*Although CALIPSO provides detailed vertical profiles of aerosols, this study does not explicitly use the height-resolved information. Instead, the aerosol occurrence is analyzed at the grid-cell level without distinguishing altitude. This approach is adopted for two main reasons. First, CALIPSO's aerosol detection is most reliable in the lower troposphere, while its sensitivity decreases significantly at higher altitudes due to signal attenuation and the difficulty of distinguishing aerosols from thin cirrus clouds (Mao et al., 2022). Therefore, focusing on overall aerosol occurrence within each grid ensures better data consistency and avoids potential misclassification errors. Second, the Ni analyzed in this study corresponds to temperatures below -30℃, the relevant aerosols are those that can influence cloud formation through vertical transport or large-scale dynamical processes, rather than being co-located at the same altitude. Hence, by integrating aerosol occurrence over the entire column within the same grid, the analysis effectively captures the overall influence of low-level dust or smoke aerosols on upper-tropospheric ice clouds, without introducing additional uncertainty from vertical matching. Therfore, the overall comparison, statistical results, and main conclusions remain robust.*

7、Page 9, line 9: Your average ICNC concentration of 187.48 L-1 is close to 150 L-1. This is a rather low difference when keeping in mind that ICNC values can be in the range from 0.1 to 10000 L-1. What about information about the standard deviations, in addition to the average values? Should be mentioned, too! Please avoid numbers with two digits after the decimal point! Instead of 187.48 L-1 it is sufficient to write 187 L-1.

Respone:Thank you for this helpful comment. The numerical values have been rounded as suggested. In addition, the manuscript has been updated to include the corresponding standard deviation information, as recommended.

8、Page 9, line 14: Agreement with observations from 0.1 to 10000 L-1 (covering 5 orders of magnitude) does not indicate any level of reliability of your data.

Respone:We agree with the reviewer that an agreement spanning several orders of magnitude does not, by itself, demonstrate the reliability of the data. Accordingly, the corresponding

**statement has been removed from the revised manuscript.**

9、Page 9,     lines 20-23: Numbers of 213 L-1, 253 L-1 and even 142 L-1 are so close together. I expected much larger, more contrasting differences between south and north. Why are the differences so low? Is that caused by all the DARDAR assumptions? Any comment?

**Respone:Thank you for this important comment. In our original analysis, the south－north contrast in ICNC was indeed much larger, as shown in the former version of Fig. 1, because the statistics were based on averaging ICNC along each individual profile. Several reviewers, however, pointed out that this profile-averaged approach is not commonly used in DARDAR-Ni climatological studies and recommended adopting a more standardized grid-based normalization method.**

**Following this advice, the revised analysis (shown in the updated Fig. 2) now first aggregates all ICNC values within each grid cell and normalizes them by the number of valid occurrences. This method effectively reduces sampling inhomogeneity between regions but also compresses the numerical contrast between south and north. As a result, the absolute differences (e.g., 213 L$^{-1}$ vs. 142 L$^{-1}$) appear smaller than in the original profile-based statistics, even though the spatial pattern—higher ICNC in the south and lower ICNC in the north—remains consistent and physically robust.**

**Therefore, the reduced difference is a consequence of the revised statistical method, not an artifact of DARDAR retrieval assumptions. The physical contrast between southern convection-dominated regions and northern INP-rich regions is still clearly reflected in the spatial distribution.**

[Figure]

Fig.1

[Figure]

Fig.2

10、Figure 1: What does it mean: The black dotted line represents a standard error of 60 L-1? At the same time, the color plot shows ICNC values of 120 to 240 L-1? The same for the other 80 and 100 L-1 standard error lines? What do you want to say with these lines? Does that indicate the variability in the DARDAR data sets, or the errors in the colored values? Please explain clearly what these error lines mean!

**Respone:Thank you for pointing out the need for clarification. In the revised manuscript, additional explanation has been added to Fig. 1 and the corresponding caption. The black contour lines represent the standard error of the ICNC values derived directly from the DARDAR-Nice retrieval uncertainty (icnc_5um_error). This information is now explicitly described in the revised figure caption and text.**

11、Page 10, line 15: This process enhances homogeneous nucleation···. thereby increasing ICNC over the southern region. How do you know? This is just your opinion (speculation). The impact of dynamics (updraft characteristics), aerosol and INP concentration levels at given temperature, and humidity conditions is rather complex. Simple conclusions are thus impossible! This should be stated!

Page 10, lines 17-30: The same here! 'Easy' and trivial solutions and conclusions cannot be obtained or drawn! The occurrence of high water vapor levels is a prerequisite for the evolution of clouds. However, without exceeding the ice-saturation-ratio threshold value for ice nucleation, nothing will happen, no ice formation will be possible. And here updraft occurrence (amplitude, speed, duration) comes into play.

**Respone:We thank the reviewer for this important comment. We agree that the processes controlling ice crystal number concentration involve a complex interplay between dynamical conditions (e.g., updraft strength and duration), aerosol and INP availability, temperature, and humidity, and that simple or definitive conclusions are not possible based on satellite**

**observations alone. In response, the relevant discussion has been substantially revised in Section 3.1.**

*During the ASM, frequent deep convection in the southern TP facilitates the transport of warm, moist air and water droplets from the Indian Ocean and the Bay of Bengal to higher altitudes (He et al., 2019). Under moist conditions, ascending air parcels are more likely to experience a prolonged period of ice supersaturation, thereby increasing the probability of exceeding the supersaturation threshold required for homogeneous ice nucleation. In humid environments, air parcels can therefore maintain supersaturated conditions for a longer duration, making homogeneous nucleation more likely to dominate under such circumstances (Zhao et al., 2018). By contrast, heterogeneous nucleation is initiated by INPs and typically requires a lower ice supersaturation threshold, allowing it to occur earlier during ascent (DeMott et al., 2010). As a result, in environments with abundant water vapor, homogeneous nucleation may gain a relative advantage in competition with heterogeneous nucleation, favouring the formation of higher $N_i$.*

*This interpretation is consistent with the relatively high $N_i$ observed over the southern TP during summer. However, within an observational framework alone, the respective contributions of dynamical conditions, aerosol properties, and thermodynamic processes cannot be fully disentangled. The interpretation presented here should therefore be regarded as a qualitative explanation based on physical consistency rather than a definitive attribution.*

*In addition to convective activity, the presence of INPs also plays a critical role in modulating $N_i$ over the TP. Zhao et al. (2018), using nine years of satellite observations, demonstrated that ice crystal formation is regulated not only by the availability of INPs but also by ambient water vapor conditions. This highlights the important role of moisture as a prerequisite for cirrus cloud evolution, while emphasizing that high water vapour availability alone is not sufficient to guarantee ice formation. Ice nucleation can only occur when the ice saturation ratio exceeds the threshold required for freezing; without reaching this threshold, no ice formation is possible (Gettelman et al., 2010). As a result, moisture should be regarded as a necessary background condition rather than a direct or sufficient driver of ice crystal formation.*

*Consequently, when investigating the relationship between INPs and $N_i$, directly comparing INPs and $N_i$ across all grid cells may lead to misleading interpretations. This is because differing atmospheric conditions, particularly variations in moisture and the development of ice supersaturation, can strongly influence whether ice formation occurs. For example, high $N_i$ in one grid cell may primarily reflect favourable moisture conditions that allow supersaturation to be achieved, rather than an enhanced influence of INPs, whereas in another grid cell the potential effect of INPs may be masked if the supersaturation threshold is not exceeded.*

*In principle, restricting the analysis to grid cells with broadly similar atmospheric conditions would allow a more direct comparison. However, the TP exhibits pronounced spatial heterogeneity, especially between its northern and southern regions. To partially account for differences in moisture-related thermodynamic conditions, this study introduces the IWC confined INPs concentration (ICIC), defined as the logarithm of the ratio between the occurrence number of smoke (or dust) particles and IWC within each grid cell (Eq. 3). By standardization, this metric improves the comparability of the analysis to some extent. To further demonstrate the robustness of this normalization, we compute the partial correlation between INPs and $N_i$ after removing the effect of IWC. The resulting coefficient, r = -0.38, confirms that the ICIC formulation effectively reduces moisture-related confounding.*

12、Page 11, line12: ··· heterogenous nucleation of INPs promotes the formation of larger crystals. How do you know? This is not a 'law'. The size of the INP reservoir and occurring strength of the updrafts (speed, length of updraft period) strongly influence the cirrus evolution. There is no easy solution available.. That should the basic message of the discussion.

**Respone:We thank the reviewer for this important comment. We fully agree that the statement suggesting heterogeneous nucleation of INPs promotes the formation of larger ice crystals should not be interpreted as a general law. As noted by the reviewer, cirrus cloud evolution is strongly influenced by multiple interacting factors, including the size and availability of the INP reservoir, as well as the strength and duration of updrafts, and no simple or universal relationship can be inferred.**

**We acknowledge that these processes cannot be robustly quantified using satellite observations alone. Accordingly, the manuscript has been revised to weaken this interpretation and to clearly indicate that it represents a preliminary, hypothesis-driven discussion rather than a definitive conclusion. Related statements have been modified or removed where necessary, and the limitations of the present study are now explicitly discussed in the final discussion section.**

Page 11, lines 17 -19: Such safe statements are welcome!

13、Page 12, line 7: Growth of ice crystals plays an important role! Ice crystals growth has also an influence on the DARDAR ICNC values. For that reason, I want see at least some comparison of ICNC solution for particle ensembles >5 and >25 micrometer.

**Response:We agree with the reviewer that ice crystal growth can influence the ICNC retrieved by the DARDAR product. In this study, the analysis is conducted at the scale of the entire Tibetan Plateau, and ICNC for particles larger than 5 μm is used as a representative measure of ice crystal number concentration at this regional and climatological scale.**

**Nevertheless, following the reviewer's suggestion, additional analyses using a particle size threshold of >25 μm have been performed. The results show that, compared with the >5 μm threshold, the use of >25 μm mainly leads to differences in the absolute ICNC values, while the main spatial patterns and qualitative conclusions remain unchanged.**

[Figure]

14、Page 12, line 14: Please repeat briefly that you assume homogeneous freezing when CALIOP does not indicate the occurrence of dust or smoke, and the aerosol type is just 'clean'.

**Respone:Thank you for this suggestion. The manuscript has been revised to explicitly restate that grid points classified as "clean".**

*In cases where CALIOP does not indicate the presence of dust or smoke and the aerosol type is classified as 'clean', ice formation is assumed to occur via homogeneous freezing.*

15、Page 13, line: ⋯below -38°C, ⋯.. and when there are no INPs!

**Respone:Thank you for pointing this out. The corresponding description has been revised to correct and clarify the conditions for homogeneous freezing.**

16、Pgae 13, line 5: Do not forget the importance of updraft speed and duration (determines how many ice crystals are produced), besides temperature.

**Respone:We agree that updraft speed and duration play an important role in determining ice crystal number concentrations. In the present study, the analysis is based on a 10-year summer climatology derived from satellite observations. At this temporal and spatial scale, differences in updraft characteristics associated with different ice formation pathways cannot be resolved and are therefore implicitly averaged in the statistical framework. As a result, the inferred Ni reflects climatological conditions rather than individual cloud-scale processes.**

17、Page 13, lines 8-13: The uncertainty in the DARDAR products is large, and the found differences in the ICNC numbers are comparably small! I do not think that it is possible to draw such clear conclusions and to make clear statements when the differences are so small? Please provide a more careful discussion! Avoid speculations!

**Respone: This issue has been carefully considered and addressed in earlier revisions.**

18、Please include the impact of gravity waves and large scale dynamics and related updraft characteristics in your discussion! Simple conclusions, as presented, are not possible. You may provide your opinion and interpretation of the observations (given in the figures). But indicate, that your argumentation is just an option, a hypothesis!

**Respone:We thank the reviewer for this important comment. In response, substantial revisions have been made throughout the discussion. In addition, the discussion has been systematically revised to avoid simple or definitive conclusions. The interpretations presented are now clearly framed as one possible explanation of the observed features, based on satellite-derived statistical relationships, rather than as proven mechanisms. The hypothesis-driven nature of the argumentation is explicitly stated in the revised manuscript.**

19、To summarize my basic opinion: Clear conclusions cannot be drawn from the different scenarios with so small ICNC differences (shown in Figures 4 and 6), in view of the large uncertainties in the DARDAR products and the large natural (atmospheric) variability in the ICNC data, linked to the complex atmospheric impact (updraft frequency, speed, period, crystal growth and sedimentation aspects, crystal collision and aggregation effects, temperature and humidity

conditions.). Strong updrafts may sometimes lead to ICNC of 500-2000 L-1. Weak updrafts may often lead to ICNC from 1-10 L-1.

**Respone: Thank you for pointing this out. We further note that this study is based on a climatological analysis of approximately ten summers of satellite observations. As a result, temporal averaging inevitably smooths extreme ICNC values that may occur during individual strong updraft events. However, the aim of this work is not to capture event-scale extremes, but to investigate regional and statistical characteristics of ICNC over the Tibetan Plateau. Within this context, we argue that the averaging does not invalidate the main qualitative conclusions.**

20、Figure 4: I am confused! The green line shows observations and the red line shows homogenous nucleation. But the red line is also based on observations! … for the aerosol type 'clean'. If I am wrong, what did I overlook? What did I miss?

**Respone:Both curves are based on satellite observations. The green line represents Ni directly retrieved from all observed conditions, whereas the red line is derived from the same observations but restricted to cases classified as "clean" aerosol conditions by CALIOP, for which homogeneous freezing is assumed.**

21、Page 14, lines 8-13: This is, to my opinion, speculation. This is your opinion. My conclusion is: The differences are not significant. The updraft impact is unkown. The authors did not observe significant differences for clean aerosol conditions (hom. Freezing) compared to dusty conditions (het. ice nucleation).

**Respone:We thank the reviewer for this comment. We acknowledge that the interpretation presented in this part of the discussion is speculative and should not be regarded as a definitive conclusion. In response, this section has been revised, and the purpose of this discussion is to provide a plausible explanatory context and to highlight potential processes that may warrant further investigation in future studies.**

22、Figure 5: My question is: What do you get in the case of ICNC for particles > 25 micrometer? How do the features change?

**Respone:Thank you for this question. Following the reviewer's suggestion, ICNC for particles larger than 25 μm has been analyzed and is now shown in the figure below. Compared with the results based on the >5 μm threshold, the >25 μm ICNC exhibits differences mainly in the absolute values, while the overall spatial distribution and the main features remain largely unchanged.**

[Figure]

23、Page 15, lines 1-18: Please provide some information about typical tropopause heights! Obviously very tropical condition prevail above the Tibetan Plateau. Please avoid numbers like 103.94 L-1, better state: 104 L-1. Strong upward motions do not only transport moist air upward but also ice crystals (outflow cirrus from dissolving convective cloud towers, often denoted as liquid-origin cirrus according to the papers of Kraemer et al.). Homogeneous freezing may occur, in addition, as a further option, not as the only option. Many statements are speculative, please indicate or emphasize the hypothetic character of your statements.

**Respone:Thank you for this detailed and constructive comment. In response, the manuscript has been revised in several respects.**

*During summer, the tropopause height over the Tibetan Plateau typically ranges from about 16 - 18 km (Sun et al., 2021), providing an important vertical reference for cirrus cloud development. The top of cirrus clouds can develop near 18 km with a relatively low Ni for the case of non-deep convection activity, and at 14 km, reaches its peak of 104 L-1. When deep convection activity occurs, the Ni at the same altitude is significantly higher, and at 14 km, reaches its peak of 162 L-1. During summer, strong upward motions over the southern TP can transport both moist air and pre-existing ice crystals from the lower troposphere into the upper troposphere via convective outflow anvils. These processes may create favorable conditions for enhanced Ni, while homogeneous nucleation may additionally occur under sufficiently cold and supersaturated conditions. It is therefore suggested that the observed Ni peak near 14 km is associated with the combined effects of convective transport, dynamical accumulation, and ice formation processes.*

24、Page 16, lines 3-11: What about a possible quick depletion of the dust INP reservoir? Do you have CALIOP data that indicate that the INP reservoir is (always) very large so that a strong decrease of INP concentration is unlikely? Huang et al. (2021) is published in Plateau Meteorlogy. Please provide further references in well known journals.

**Respone:Thank you for this comment. The potential limitation related to a possible rapid depletion of the dust INP reservoir has been acknowledged and discussed in the final section of the manuscript. In addition, new references from well-established international journals have**

**been added in the revised manuscript to support the discussion.**

25、Figure 7b: How are the profiles in Fig. 7b computed? Is that the average of all profiles used in Fig. 7a?

**Respone:Thank you for this question. Additional clarification has been added in Section 3.2.3 of the revised manuscript.**

*In the northern part of the TP, convective activity is relatively weak, but dust aerosol content is high (Fig. 7a). The increase in Ni is primarily attributed to heterogeneous nucleation induced by INPs. Considering the frequent dust activity in this region, we selected grid points with ICIC(dust) greater than -5 as the primary study area. These grid points are predominantly located in the northern Plateau, adjacent to Xinjiang, a typical semi-arid region characterized by abundant dust aerosols. These dust particles facilitate water vapor adsorption in the lower atmosphere and promote ice crystal formation through heterogeneous nucleation (Hoose and Möhler, 2012; Huang et al., 2021).*

26、Page 16 lines 12-27: Now potential sedimentation contributions to ICNC at different height levels come into play, for the first time in this manuscript. The paragraph is full of hypotheses and opinion-like statements. Please clearly indicate the hypothetic character of your statements. The atmospheric conditions, processes, and impacts are too complex to allow simple and straight forward conclusions and to provide the impression: We tell you the truth! I leave out here to repeat my 'warnings' already stated above⋯. We need an open discussion, hypotheses are welcome, but we need to avoid the impression that we found clear results (answers) showing in detail and very clear how ice crystals formed, either via homogeneous or heterogeneous ice nucleation path ways, and what the role of aerosols, temperatures and water vapor is in these ice formation processes.

**Respone:Thank you for this comment. In response to the reviewer's concerns, the corresponding part of the manuscript has been revised to clearly indicate the hypothetical nature of the discussion and to avoid overly definitive conclusions.**

*Fig. 7b illustrates the effect of dust aerosol particles on Ni in this area. The results suggest that the presence of dust is associated with a reduction of Ni in cirrus clouds, with concentrations above 12 km becoming very small. During non-dust periods, although INPs remain present, their activation efficiency may be relatively high, allowing a large fraction of aerosol ice nuclei to be activated, resulting in weaker suppression of ice crystal formation. In contrast, elevated dust concentrations in the lower atmosphere may enhance heterogeneous nucleation, thereby consuming available water vapor and potentially inhibiting additional ice crystal formation, which could lead to a reduction of Ni. Within this interpretative framework, Ni during non-dust periods tends to be higher than during dust conditions. However, due to limited water vapor in this region, a large fraction of moisture may already be depleted in the lower atmosphere, which could contribute to very low Ni above 12 km. Consequently, the suppressive effect of heterogeneous nucleation may limit ice crystal formation through homogeneous nucleation at higher altitudes, making cirrus cloud development more difficult in these upper layers. In regions with low water vapor content, INPs may play an important role in modulating Ni.*

27、Page 17, line 5 to page 18, line 6: The paragraphs are again full of speculative statements.

Again, we need an improved scientific discussion with clear indication that hypotheses are given. Speculations need to be indicated as such! Avoid speculation as much as possible!

**Respone:Thank you for this comment. In response to the reviewer's concerns, the corresponding part of the manuscript has been revised to clearly indicate the hypothetical nature of the discussion and to avoid overly definitive conclusions.**

*Besides dust aerosol, smoke aerosol particles generated by human activities are considered another potential source of heterogeneous nucleation for cirrus clouds over the TP. In this research, grid points with ICIC(smoke) greater than −6.5 were selected as the primary research region to examine the possible influence of smoke INPs on Ni (Fig. 8a).*

*It is observed that the presence of smoke aerosols is associated with a decrease in Ni, with the maximum vertical extent of cirrus clouds limited to around 14 km (Fig. 8b). One possible interpretation is that, during smoke events, ice crystal formation in the lower atmosphere may be influenced by smoke-derived INPs, under which heterogeneous nucleation could become more active. The relatively high abundance of smoke INPs may enhance competition among ice particles, potentially suppressing additional ice crystal formation and resulting in lower Ni compared to non-smoke conditions. In this hypothetical framework, smoke INPs may be efficiently activated through heterogeneous nucleation, while any remaining water vapor could still contribute to ice formation via homogeneous nucleation. From this perspective, Ni tends to be higher during non-smoke periods, when fewer INPs may lead to weaker suppression effects. However, due to the inherently low water vapor content in this region, the vertical development of cirrus clouds appears to be constrained, and even during non-smoke periods, the maximum cloud height remains limited to approximately 17 km.*

*Homogeneous nucleation is often associated with a peak in Ni near 14 km. However, under dust and smoke conditions, such a peak is not clearly observed. One possible explanation is that dust and smoke aerosols are mainly concentrated over the northern TP, where the vertical wind speed around 400 hPa is relatively weak (Fig. 6b). Reduced vertical transport may limit the upward redistribution of ice crystals, thereby influencing the vertical structure of Ni.*

28、Some remaining questions: How is smoke identified by using CALIOP backscatter and depolarization ratio information? As I know, smoke particles may be non-spherical when transported quickly into the upper troposphere but in the lower and middle free troposphere they are spherical and do not cause depolarization. In contrast, dust always shows high depolarization ratios! How can we unambiguously distinguish smoke from dust?

**Respone: In this study, smoke and dust aerosols are not identified using independent thresholds of CALIOP backscatter or depolarization ratio. Instead, aerosol types are taken directly from the CALIPSO Level-2 aerosol classification product.**

29、Figure 7b: What do you mean with 'non-dust'? Do you assume 'clean' conditions (homogeneous freezing conditions) or 'clean plus smoke' conditions?

**Respone:In Fig. 7b, "non-dust" refers to conditions in which no dust aerosol is detected in the CALIPSO classification. These cases may include clean conditions as well as situations influenced by other aerosol types (e.g., smoke), but they are grouped together solely based on the absence of dust.**

30、Figure 8b: The same here! What do you mean with 'non-smoke'? Do you assume 'clean' conditions (homogeneous freezing conditions) or 'clean plus dust' conditions?
**Respone:Same as the response above, , but applied to smoke aerosols.**

31、At the end, the abstract and the conclusion sections need to be carefully updated when the revision of the main parts of the manuscript is completed.
**Respone:Thank you for this comment. In accordance with the reviewer's suggestion, both the abstract and the conclusion sections have been carefully revised to ensure consistency with the updated discussion and the revised main body of the manuscript.**

Without significant improvement of the manuscript with my comment statement as a guide, I will not recommend to publish this study.

**Comments from Anonymous Referee4**

This manuscript presents a climatological analysis of cirrus cloud properties over the Tibetan Plateau, with a particular focus on ice crystal number concentration and its relationship to aerosols and different ice nucleation processes. The topic is scientifically relevant and clearly within the scope of ACP, and the overall methodological approach is interesting and merits further investigation. However, in its current form, the manuscript has several important limitations. In particular, some of the conclusions appear stronger than what can be robustly supported by the analyses presented.

I therefore do not recommend acceptance of the manuscript in its present state and suggest major revisions. The authors would need to strengthen and better document their analyses, clarify aspects of the methodology, and refine or more carefully justify several of their interpretations in order to support their conclusions. The general and specific comments below focus on the most significant issues identified in the manuscript.

General comments:

This manuscript makes strong claims regarding the impact of aerosols on cirrus clouds and the competition between homogeneous and heterogeneous ice nucleation. While I agree in principle that such effects can, and should, be observable to some extent, I do not find that the present study sufficiently supports these claims. First, the ICIC metric is somewhat unclear and would benefit from a clearer justification (see below). More importantly, comparisons between regions with differing aerosol concentrations may also implicitly involve comparisons between different meteorological regimes, which could explain at least in part the observed differences in Ni. This issue may be particularly relevant over the Tibetan Plateau, given the strong geographical and dynamical contrasts within the region. This point is partly acknowledged in Section 3.2.3, where the presence of dust and smoke is associated with weak vertical winds conditions. The manuscript would benefit from a more convincing effort to disentangle meteorological influences from aerosol effects, and at a minimum from explicitly acknowledging that the observed differences in the satellite data may not be attributable to aerosol presence alone.

I find the discussion of homogeneous nucleation to lack precision and, at times, to be misleading. On several occasions (see below), the manuscript refers to the "homogeneous nucleation of water vapour," which is not physically possible in the atmosphere and therefore requires clarification. More generally, it is unclear throughout the manuscript whether the authors are referring to the homogeneous freezing of supercooled water droplets (particularly relevant in deep convective regions) or to the in-situ homogeneous freezing of aqueous aerosols under sufficiently high supersaturation (also relevant in this region). The manuscript would benefit from a more specific and physically grounded discussion of the ice formation processes being considered. In addition, clarity would be in my opinion substantially improved by explicitly distinguishing between in-situ formed ice and liquid-origin ice in the discussions.

Several datasets are used simultaneously in the analysis. While each of them appears reasonable and of good quality, they have different sensitivities and retrieval characteristics (e.g., Ni from combined lidar-radar retrievals, IWC from radar-only retrievals, and aerosol classification from lidar-only observations). These differences could substantially influence the results and their interpretation, and the associated limitations should be discussed in more detail.

Specifically, the aerosol product is not sufficiently described. It would be helpful to specify the exact CALIPSO aerosol product used (product name, version, and horizontal/vertical resolution), and to clarify whether the aerosol information was temporally and vertically colocated with the cirrus observations, or instead whether any aerosol detected within the full column and/or within a given grid cell was considered. In addition, it would be useful to clarify whether aerosols are expected to be reliably detectable in the vicinity of deep convective events with the chosen approach, and to discuss the implications and potential limitations of the method used.

The ICIC (IWC confined INPs concentration) variable is also unclear. It is described as "the logarithm of the ratio between the number of smoke (dust) particle occurrences and the IWC in each grid cell", with no further justification. Is there a reference for this technique, or at least provide a justification as to what

**Respone: We thank the reviewer for the thorough and constructive general comments. We acknowledge that the original manuscript contained overly strong statements regarding aerosol impacts on cirrus clouds and the competition between homogeneous and heterogeneous ice nucleation, which were not always sufficiently supported by the satellite-based analyses. In response, the manuscript has undergone substantial revision to weaken definitive claims and to clearly frame the interpretations as hypothesis-driven and statistical in nature.**

**With respect to the attribution of observed Ni differences, we agree that variations in aerosol conditions are often intertwined with differences in meteorological regimes, particularly over the Tibetan Plateau where strong geographical and dynamical contrasts exist. The revised manuscript now more explicitly acknowledges that the observed differences in Ni cannot be attributed to aerosol presence alone, and that dynamical and thermodynamic factors play an important role.**

**We also agree that the discussion of homogeneous nucleation required clarification and improved physical precision. All references to the "homogeneous nucleation of water vapour" have been corrected. The revised manuscript now clearly distinguishes between homogeneous freezing of supercooled water droplets (liquid-origin ice) and in-situ homogeneous freezing of aqueous aerosols under high ice supersaturation. In addition, in-situ formed ice and**

liquid-origin ice are explicitly distinguished throughout the discussion to improve physical clarity.

The use of multiple satellite datasets with different sensitivities and retrieval characteristics has been clarified, and the associated uncertainties and limitations are now discussed in more detail. In particular, the CALIPSO aerosol product used in this study is now more clearly described, including product type, version, and resolution. The methodology for combining aerosol and cirrus information has been clarified to emphasize that aerosol occurrence is used in a climatological, grid-cell－based sense rather than through instantaneous vertical collocation with cirrus observations.

Finally, the definition and interpretation of the ICIC metric have been clarified. Additional explanation has been added to justify its formulation as a statistical indicator linking aerosol occurrence and IWC at the grid-cell level, and its limitations are now explicitly discussed.

Overall, the revised manuscript places stronger emphasis on uncertainty, complexity, and the exploratory nature of the analysis, and avoids presenting the results as definitive evidence of specific ice nucleation pathways. The interpretations are now consistently framed as one possible explanation of the observed statistical patterns, intended to provide context and motivation for future, more process-oriented studies.

Specific comments :

1、Page 3 lines 22-24: This statement is misleading. Homogeneous nucleation of water vapour does not occur in the atmosphere. The wording should be revised accordingly.

**Respone: Thank you for pointing this out. The wording has been revised accordingly to correct the misleading statement, and the reference to homogeneous nucleation of water vapour has been removed from the manuscript.**

*The homogeneous freezing of supercooled water droplets or aqueous aerosol particles to form ice crystals requires temperatures below approximately $-38\,^{\circ}\!C$ and sufficiently high ice supersaturation (Duft and Leisner, 2004; Murray et al., 2010).*

2、Page 5 line 4: Please specify more precisely which months are included in the definition of "summer." In addition, please clarify whether both daytime and nighttime measurements are used. If so, it would be important to discuss whether the absence of nighttime CALIPSO retrievals after 2012 has an impact on the analysis and results.

**Respone: Thank you for this comment. In the revised manuscript, the definition of "summer" has been clarified by explicitly specifying the months included in the analysis. Both daytime and nighttime CALIPSO measurements are used in this study(2.1). We have carefully checked the CALIPSO product documentation and data availability, and no evident loss of nighttime observations after 2012 was found for the aerosol product used in this study. The data completeness is consistent with the information provided in the official CALIPSO documentation.**

*This study uses ten summers (June-July-August, JJA) of multi-satellite observations during 2006 to 2016*

*In this study, both daytime and nighttime satellite observations are included, the aerosol*

*information is used to characterize climatological, grid-cell-averaged aerosol occurrence rather than instantaneous cloud-aerosol collocation.*

3、Page 8 lines 6-8: As noted above, homogeneous freezing near -38°C applies to supercooled water droplets and aqueous aerosols, not to water vapour itself. This sentence should be corrected accordingly to avoid confusion about the underlying physical process.

**Respone: Thank you for this comment. The sentence has been revised accordingly to clarify that homogeneous freezing near −38 °C refers to supercooled water droplets and aqueous aerosols, rather than water vapour, thereby avoiding confusion about the underlying physical process.**

*It is generally acknowledged that temperatures near −38 °C represent the threshold for homogeneous freezing of supercooled water droplets and aqueous aerosol particles under sufficiently high ice supersaturation (Koop and Murray, 2016; Duft and Leisner, 2004; Murray et al., 2010).*

4、Page 8 lines 9-12: "However, due to the continuous dynamic growth of ice particles through condensation, accurate simulation remains challenging. Moreover, purely homogeneous nucleation events are extremely rare in the natural atmosphere." - This statement would benefit from further clarification and appropriate references. In particular, the claim that purely homogeneous nucleation events are extremely rare is not self-evident in the context of this study, given that the manuscript aims to identify signatures of homogeneous freezing from CALIPSO observations. Additional justification and supporting literature should be provided.

**Respone:Thank you for this comment. In response, the corresponding statement has been revised to improve clarity, and additional explanation and supporting references have been added to the manuscript.**

*Moreover, classical nucleation theory suggests that ice formation under purely homogeneous freezing conditions is generally considered to be uncommon in the natural atmosphere (Maeda, 2021).*

5、Page 10 lines 26-27: "For example, high Ni in one grid cell could be due to abundant water vapor rather than the effect of INPs" - Please clarify what is meant by this statement. Specifically, does "abundant water vapor" refer to higher supersaturation leading to enhanced in-situ ice formation, or to the freezing of water droplets within deep convective updrafts (i.e. liquid-origin ice)? Clarifying this point would help to better interpret the role attributed to INPs.

**Respone: Thank you for this comment. The corresponding part of the manuscript has been revised to clarify the intended meaning.**

*In addition to convective activity, the presence of INPs also plays a critical role in modulating Ni over the TP. Zhao et al. (2018), using nine years of satellite observations, demonstrated that ice crystal formation is regulated not only by the availability of INPs but also by ambient water vapor conditions. This highlights the important role of moisture as a prerequisite for cirrus cloud evolution, while emphasizing that high water vapour availability alone is not sufficient to guarantee ice formation. Ice nucleation can only occur when the ice saturation ratio exceeds the threshold required for freezing; without reaching this threshold, no ice formation is possible (Gettelman et al., 2010). As a result, moisture should be regarded as a necessary background*

*condition rather than a direct or sufficient driver of ice crystal formation.*

*Consequently, when investigating the relationship between INPs and Ni, directly comparing INPs and Ni across all grid cells may lead to misleading interpretations. This is because differing atmospheric conditions, particularly variations in moisture and the development of ice supersaturation, can strongly influence whether ice formation occurs. For example, high Ni in one grid cell may primarily reflect favourable moisture conditions that allow supersaturation to be achieved, rather than an enhanced influence of INPs, whereas in another grid cell the potential effect of INPs may be masked if the supersaturation threshold is not exceeded.*

*In principle, restricting the analysis to grid cells with broadly similar atmospheric conditions would allow a more direct comparison. However, the TP exhibits pronounced spatial heterogeneity, especially between its northern and southern regions. To partially account for differences in moisture-related thermodynamic conditions, this study introduces the IWC confined INPs concentration (ICIC), defined as the logarithm of the ratio between the occurrence number of smoke (or dust) particles and IWC within each grid cell (Eq. 3). By standardization, this metric improves the comparability of the analysis to some extent. To further demonstrate the robustness of this normalization, we compute the partial correlation between INPs and Ni after removing the effect of IWC. The resulting coefficient, r = -0.38, confirms that the ICIC formulation effectively reduces moisture-related confounding.*

6、Section 3.2.1 and Figure 4: It would be helpful to explain more clearly how "observations" and "homogeneous nucleation" are distinguished using the satellite dataset, as the basis for this separation is currently difficult to follow. In addition, the interpretation that lower Ni reflects heterogeneous nucleation suppressing homogeneous nucleation would benefit from further justification, since alternative explanations also seem plausible. For instance, lower Ni could reflect weaker or less frequent updrafts (and therefore lower supersaturation), or differences in cloud origin (e.g., predominantly in-situ cirrus with limited contribution from liquid-origin ice detrained from deep convective updrafts).

**Respone: Thank you for this comment. The corresponding description in Section 3.2.1 and the discussion related to Figure 4 have been revised to more clearly explain the distinction between observations and homogeneous nucleation inferred from the satellite data. In the revised manuscript, "observations" refer to ice crystal number concentrations directly retrieved from satellite measurements, while "homogeneous nucleation" refers to the ice crystal number concentrations under clean aerosol conditions. In addition, the interpretation has been adjusted to acknowledge alternative explanations, and overly definitive statements have been weakened accordingly.**

*Ni for each vertical layer is calculated using Eq. (2), and Fig. 4 depicts the vertical distribution of the Ni from satellite observations and homogeneous nucleation. In cases where CALIOP does not indicate the presence of dust or smoke and the aerosol type is classified as 'clean', ice formation is assumed to occur via homogeneous freezing.*

*It is widely accepted that the formation of larger ice crystals through heterogeneous nucleation processes takes precedence over homogeneous nucleation (Shi et al., 2017; Barahona and Nenes, 2009). In fact, heterogeneous nucleation is the dominant ice formation mechanism at*

*temperatures above -38℃, whereas homogeneous nucleation occurs only when the temperature drops below -38℃ and when there are no INPs. Although homogeneous nucleation is the major contributor to the Ni (Cantrell and Heymsfield, 2005), heterogeneous nucleation has lower activation requirements and may occur earlier, potentially consuming water vapor and influencing subsequent ice formation. Under this interpretation, the observed Ni being lower than that expected under conditions favorable for homogeneous freezing could be consistent with the influence of heterogeneous nucleation. However, alternative explanations cannot be excluded. For example, lower Ni may also reflect weaker or less frequent updrafts, which would limit the development of high ice supersaturation, or differences in cloud origin, such as a predominance of in-situ cirrus with limited contribution from liquid-origin ice detrained from deep convective updrafts (Lyu et al., 2025;Gryspeerdt et al., 2018).*

7、 The inferred influence of aerosols on cirrus clouds in Figure 5 currently appears rather speculative. It would be helpful to further analyse and quantify the relationships shown in the figure to better support the proposed interpretation, potentially by combining the cirrus properties with the aerosol information (e.g., aerosol occurrence/classification) in a more explicit way.

**Respone: Thank you for this comment. We acknowledge that the original analysis of Figure 5 had mistakes. The analysis and corresponding discussion have now been revised in the manuscript.**

*Together, these zonal and meridional distributions reveal a consistent vertical structure, with peak Ni occurring near 14 km, which could be influenced by homogeneous nucleation processes that dominate at these altitudes (Fig. 4). In contrast, Ni exhibits significant variability across both latitudinal and longitudinal directions, which may be related to the spatial distribution of water vapor and certain meteorological factors, such as vertical wind velocity, providing a foundation for the subsequent analysis in this study.*